# Chronic Glaucoma Using Biodegradable Microspheres to Induce Intraocular Pressure Elevation. Six-Month Follow-Up

**DOI:** 10.3390/biomedicines9060682

**Published:** 2021-06-16

**Authors:** Maria Jesus Rodrigo, David Garcia-Herranz, Manuel Subias, Teresa Martinez-Rincón, Silvia Mendez-Martínez, Irene Bravo-Osuna, Ana Carretero, Jesús Ruberte, Julián Garcia-Feijoo, Luis Emilio Pablo, Rocío Herrero-Vanrell, Elena Garcia-Martin

**Affiliations:** 1Miguel Servet Ophthalmology Research Group (GIMSO), Aragon Health Research Institute (IIS Aragon), University of Zaragoza, 50009 Zaragoza, Spain; mariajesusrodrigo@hotmail.es (M.J.R.); manusubias@gmail.com (M.S.); teresamrincon@gmail.com (T.M.-R.); oftalmosmm@gmail.com (S.M.-M.); lpablo@unizar.es (L.E.P.); 2National Ocular Pathology Network (OFTARED), Carlos III Health Institute, 28040 Madrid, Spain; davgar07@ucm.es (D.G.-H.); ibravo@ucm.es (I.B.-O.); ana.carretero@uab.cat (A.C.); jesus.ruberte@uab.cat (J.R.); jgarciafeijoo@hotmail.com (J.G.-F.); rociohv@farm.ucm.es (R.H.-V.); 3Innovation, Therapy and Pharmaceutical Development in Ophthalmology (InnOftal) Research Group, Department of Pharmaceutics and Food Technology, Health Research Institute of the San Carlos Clinical Hospital (IdISSC), 28040 Madrid, Spain; 4Center for Animal Biotechnology and Gene Therapy (CBATEG), School of Veterinary Medicine, Universitat Autònoma de Barcelona, 08193 Bellaterra, Spain; 5Department of Ophthalmology, San Carlos Clinical Hospital, Health Research Institute of the San Carlos Clinical Hospital (IdISSC), 28040 Madrid, Spain

**Keywords:** glaucoma, animal model, microspheres, neurodegeneration

## Abstract

Background: To compare two prolonged animal models of glaucoma over 24 weeks of follow-up. A novel pre-trabecular model of chronic glaucoma was achieved by injection of biodegradable poly lactic-co-glycolic acid (PLGA) microspheres (10–20 µm) (Ms20/10) into the ocular anterior chamber to progressively increase ocular hypertension (OHT). Methods: Rat right eyes were injected to induce OHT: 50% received a suspension of Ms20/10 in the anterior chamber at 0, 2, 4, 8, 12, 16 and 20 weeks, and the other 50% received a sclerosing episcleral vein injection biweekly (EPIm). Ophthalmological clinical signs, intraocular pressure (IOP), neuroretinal functionality measured by electroretinography (ERG), and structural analysis of the retina, retinal nerve fiber layer (RNFL) and ganglion cell layer (GCL) protocols using optical coherence tomography (OCT) and histological exams were performed. Results: Both models showed progressive neuroretinal degeneration (*p* < 0.05), and contralateral eye affectation. The Ms20/10 model showed a more progressive increase in IOP and better preservation of ocular surface. Although no statistical differences were found between models, the EPIm showed a tendency to produce thicker retinal and thinner GCL thicknesses, slower latency and smaller amplitude as measured using ERG, and more aggressive disturbances in retinal histology. In both models, while the GCL showed the greatest percentage loss of thickness, the RNFL showed the greatest and earliest rate of thickness loss. Conclusions: The intracameral model with biodegradable microspheres resulted more like the conditions observed in humans. It was obtained by a less-aggressive mechanism, which allows for adequate study of the pathology over longer periods.

## 1. Introduction

Glaucoma is a leading cause of blindness worldwide, with an estimated 60 million people currently suffering from it and 118.1 million forecast to be affected by 2040 [1,2]. It is characterized by the progressive death of retinal ganglion cells (RGC) and glaucomatous optic disc neuropathy, which lead to irreversible visual field alteration [3,4]. An increase in intraocular pressure (IOP) is a major risk factor strongly associated with the onset and progression of the disease [5].

In order to understand its pathogenesis, several animal models have been established to determine the causes of RGC death, including toxicity by glutamate [6,7], nitrogen oxide [8], oxidative stress or organelle dysfunction [9,10]. Genetic models, such as inherited DBA/2J for glaucoma [11,12], or induced ocular hypertension (OHT) using several pre/post trabecular mechanisms [13] based on injection of microbeads into the anterior chamber, laser induction, or sclerosing injections in the episcleral veins, have demonstrated an increase in IOP with retinotopic RGC death and axonal damage [14,15]. However, in all these models the increase in IOP is abrupt and/or short-lived. Several researchers have worked to simulate chronic glaucoma in order to understand and properly treat the pathology [14,15,16,17]. In this sense, a modified animal model of glaucoma based on intracameral injection of polystyrene microbeads (15 microns) able to induce a long-lasting increase in IOP was recently proposed. However, this model also had some drawbacks [18], such as corneal opacity, bleeding, or retinal cell death similar to retinal vein occlusion by intense IOP elevation, all very unlike real glaucoma [19]. Furthermore, all these models rapidly produce severe damage that renders neuroprotective or hypotensive drugs ineffective for testing in either the earliest phases of the disease or for prolonged periods of time. In this scenario, creating a model that produces a progressive IOP increase with chronic neuroretinal degeneration could be useful for the prospective study of glaucomatous pathology. It might also be helpful in evaluating the neuroprotection and/or regeneration produced by neuroprotective active compounds alone or, particularly, when they are included in controlled drug delivery devices. Development of an animal model of chronic glaucoma based on provoking a progressive and sustained increase in IOP might allow for time-by-time and step-by-step evaluation throughout the neurodegeneration pathway.

For this purpose, we created a modified chronic glaucoma model through intracameral injection of biodegradable poly (co-lactic-glycolic acid) (PLGA) microspheres by using two different size fractions (10–20 µm and 20–38 µm). The 10–20 µm (Ms20/10 model) fraction was selected as the optimal range to promote a progressive and sustained increase in rat eye IOP over 8 weeks of follow-up [20]. This pre-trabecular model was compared with that produced by sclerosis of the aqueous humor outflow pathway [21].

As the main objective of an animal model of glaucoma is to achieve a slow and very prolonged degeneration of the neuroretina, the comparison with acute models based on optic nerve crushing that produce complete degeneration of the retina in a few days was ruled out and other somewhat more progressive models, such as the intracameral injection of microbeads were also discarded because of the difficulties associated with in vivo monitoring of the retina due to obstruction of the visual axis (inflammatory sinechiae in latex or poliestirene beads) and to avoid the use of magnetic fields (in ferromagnetic beads) that are unlike real pathogenesis of glaucoma. Thus, the pre-trabecular animal model was compared with an animal model of hypertension that, according to the scientific literature, would allow a slow progression of the neuronal damage. Therefore, it was decided to opt for the episcleral vein sclerosis model; which had been shown to be able to maintain a constant high intraocular pressure for longer periods of time through successive injections of hypersaline solution with the consequent slow alteration of the retina. In this way, long-term pre- and post-trabecular mechanisms were compared. The model based on intracameral injection of PLGA microspheres caused neurodegeneration of the retina and optic nerve comparable to the episcleral sclerosis model (EPIm) and was less aggressive. This paper now employs both the Ms20/10 model and the EPIm to study chronic glaucomatous degeneration during a longer term (6 months of follow-up).

## 2. Materials and Methods

### 2.1. Methods, Intervention, or Testing

An interventionist and longitudinal animal study was conducted to create a chronic glaucoma model, achieved by injecting PLGA Ms into the anterior chamber of rat eyes, in order to study this common pathology.

Ms20/10 preparation was as described in [20]. The PLGA (D,L-lactide-co-glycolide) (50:50; inherent viscosity: 0.16–0.24 dL/g; Evonik España, Granollers, Spain) Ms were obtained using the oil-in-water (O/W) emulsion solvent extraction-evaporation technique. Briefly, 400 mg of PLGA was dissolved in 2 mL of methylene chloride. This organic polymer solution was emulsified with 5 mL of PVA Milli-Q water solution (1% *w/v*) (PVA; 67,000 g/mol; Merck KgaA, Darmstadt, Germany) using a homogenizer (Polytron^®^RECO, Kinematica, GmbHT PT3000, Lucerne, Switzerland) set at 7000 rpm for 1 min. The emulsion formed was poured into 100 mL of PVA Milli-Q water solution (0.1% *w/v*) and magnetically stirred for 3 h to allow Ms maturation by organic solvent extraction and evaporation. Once formed, the Ms were washed with Milli-Q water and the 20–10 µm size fraction was isolated by sieving. After that, the selected size fraction was freeze-dried (freezing: −60 °C/15 min; drying: −60 °C/12 h/0.1 mBar) and stored at −30 °C in dry conditions until use (production yield 39.37 ± 1.75%). According to the dual light scattering technique (Microtrac^®^ S3500 Series Particle Size Analyzer, Montgomeryville, PA, USA), a mean particle size of 14.07 ± 1.07 µm was observed. Scanning electron microscopy (Jeol, JSM-6335F, Tokyo, Japan) pictures showed spherical, non-porous, smooth-surfaced particles.

### 2.2. Animal Welfare and Anesthesia

All work with animals was carried out in the experimental surgery department of the Biomedical Research Center of Aragon (CIBA). The experiment was previously approved by the Ethics Committee for Animal Research (PI34/17) and carried out in strict accordance with the ARVO (Association for Research in Vision and Ophthalmology) Statement on the Use of Animals in Ophthalmic and Vision Research.

The study was conducted on 50 (40% males, 60% females) Long–Evans rats at 4 weeks of age. Rat weight at the start of the study ranged from 50–100 g. The animals were housed in standard cages with access to water and food ad libitum in rooms with 12 h dark–light cycles and controlled environments (temperature: 22 °C; relative humidity: 55%). IOP measurements and OHT injections were performed under gas anesthesia using a mixture of 3% sevoflurane gas and 1.5% oxygen. General anesthesia by intraperitoneal injections (60 mg/kg of ketamine + 0.25 mg/kg dexmedetomidine) was used for electroretinogram (ERG) and optical coherence tomography (OCT), and mydriatic eye drops containing 10 mg/mL of tropicamide and 100 mg/mL of phenylephrine (Alcon Cusí^®^ SA, Barcelona, Spain) were applied. Surgical and sedative procedures were performed in a temperature-controlled environment using warm pads, antisepsis conditions, topical anesthesia with 1 mg/mL of tetracaine + 4 mg/mL of oxibuprocaine (Anestesico doble Colircusi^®^, Alcon Cusí^®^ SA, Barcelona, Spain), antibiotic conditions with 3 mg/mL of ofloxacin (Exocin Colircusi^®^, Alcon Cusí^®^ SA, Barcelona, Spain), and eye drops and eye lubrication with 2% hypromellose (Methocel^®^ OmniVision, Munich, Germany) and 5 mg/g of erythromycin (Oftalmolosa Cusí^®^ eritromicina, Alcon Cusí^®^ SA, Barcelona, Spain). After the procedures, the animals were left to recover in an enriched oxygen atmosphere (2.5%).

### 2.3. Injection Procedure for Induced Ocular Hypertension

Twenty-five rats received a 2 µL [20,22] PLGA Ms20/10 (10% *w/v*) suspension injected into the anterior chamber of the right eye (RE) using a Hamilton^®^ syringe and glass micropipette via corneal superotemporal puncture, applied at baseline and biweekly for the first month and then once monthly until week 20. Thus, a total of 7 intraocular injections were performed in the study (Figure 1). The EPIm was applied to the RE of the other 25 rats as per Morrison et al. [21], with biweekly injections performed if IOP was under 20 mmHg. All injections were performed by the same researcher.

### 2.4. Clinical Signs and Intraocular Pressure Measurements

Ophthalmological clinical signs such as ocular hyperemia, cornea alteration, infection or intraocular inflammation were evaluated weekly, as was IOP measured with a Tonolab^®^ rebound tonometer (Tonolab, Tiolat Oy, Helsinki, Finland) in the morning to avoid the fluctuation pattern due to the circadian cycle [23]. Both right and left eyes were measured (always measuring the right eye first) and sexes were randomly selected with a sex ratio 40%/60% male/female at every time point. The IOP value was the average of three consecutive measurements, taken from the average of 6 rebounds. Ocular hypertension was considered when intraocular pressure levels were greater than 20 mmHg.

### 2.5. Neuroretinal Examination

#### 2.5.1. In Vivo Optical Coherence Tomography

Neuroretinal structures were studied with the Spectralis OCT device (Heidelberg^®^ Engineering, Heidelberg, Germany) and with a contact lens adapted to the rat cornea to obtain higher-quality recordings. It was performed at baseline and at weeks 8, 12, 18 and 24 after the start of the study, as earlier behavior had been evaluated in our previous paper [20]. Automatic segmentation protocols such as retina posterior pole (R), retinal nerve fiber layer (RNFL) and ganglion cell layer (GCL) were used. These protocols analyzed an area of 3 mm around the center of the optic disc by performing 61 b-scans. Subsequent follow-up examinations were conducted at this same location using the eye-tracking software and follow-up application. The retina and GCL were analyzed by mimicking the 9 ETDRS (early treatment diabetic retinopathy study) areas [24], which included a central (C) 1 mm circle centered on the optic disc (although no fovea exists in rats), and inner (inferior: II; superior: IS; nasal: IN; temporal: IT) and outer (inferior: OI; superior: OS; nasal: ON; temporal: OT) rings measuring 2 and 3 mm in diameter, respectively, as well as total volume (TV). The RNFL protocol provides measurements of the 6 peripapillary sectors (inferotemporal: IT; inferonasal: IN; superotemporal: ST; superonasal: SN; nasal: N; and temporal: T). Retinal thickness comprises the area from the inner limiting membrane to the retinal pigment epithelium; the RNFL comprises the area from the inner limiting membrane to the GCL boundaries; and the GCL comprises the area from the RNFL to the inner nuclear layer boundaries.

#### 2.5.2. In Vivo Electroretinography

Neuroretinal structure functionality was studied with electroretinography (ERG) (Roland consult^®^ RETIanimal ERG, Brandenburg, Germany) using the flash scotopic ERG and photopic negative response (PhNR) protocols at baseline, week 12 and week 24. To perform the scotopic ERG test, animals were dark-adapted for 12 h and their pupils fully dilated. Active electrodes were placed on the cornea, references were placed at both sides under the skin, and the ground electrode was placed near the tail. Acceptable electrode impedance was considered to be <2 kΩ between electrodes. Both eyes were simultaneously tested with a Ganzfeld Q450 SC sphere employing white light-emitting diode (LED) flashes for stimuli. Seven steps were performed at increasing intervals and intensity of luminance (response to dim stimulus in dark adaptation to evaluate rod response; step 1: −40 dB, 0.0003 cds/m^2^, 0.2 Hz [20 recordings averaged]; step 2: −30 dB, 0.003 cds/m^2^, 0.125 Hz [18 recordings averaged]; step 3: −20 dB, 0.03 cds/m^2^, 8.929 Hz [14 recordings averaged]; step 4: −20 dB, 0.03 cds/m^2^, 0.111 Hz [15 recordings averaged]; step 5: −10 dB, 0.3 cds/m^2^, 0.077 Hz [15 recordings averaged]; response to bright stimulus in dark adaptation to analyze mixed rod-cone response; step 6: 0 dB, 3.0 cds/m^2^, 0.067 Hz [12 recordings averaged]; and oscillatory potentials; step 7: 0 dB, 3.0 cds/m^2^, 29.412 Hz [10 recordings averaged]). The PhNR protocol was performed after light adaptation to a blue background (470 nm, 25 cds/m^2^), and a red LED flash (625 nm, −10 dB, 0.30 cds/m^2^, 1.199 Hz [20 recordings averaged]) was used as stimuli. Latency (in milliseconds) and amplitude (in microvolts) were studied in a, b and PhNR waves.

Biased examinations were rejected or manually corrected if the algorithm had obviously failed.

#### 2.5.3. Histology

The animals were humanely euthanized with an intracardiac injection of sodium tiopenthal (25 mg/mL) under general anesthesia and their eyes were immediately enucleated.

Paraffin-embedded eyes were sectioned (3 µm) along the eye axis, deparaffinized and rehydrated. After several washes in phosphate buffered saline (PBS), sections were incubated overnight at 4 °C with the following primary antibodies: rabbit anti-mouse glial fibrillary acidic protein (GFAP) (DAKO, Glostrup, Denmark) at 1:1000 dilution; rabbit anti-mouse glutamine synthetase (GS) (Sigma-Aldrich, St Louis, MO, USA) at 1:500 dilution; goat anti-human Iba1 (Abcam, Cambridge, UK) at 1:100 dilution; mouse anti-mouse PKCα (Sigma-Aldrich,) at 1:100 dilution; and goat anti-mouse cone arrestin (Santa Cruz Biotechnology, Heidelberg, Germany) at 1:100 dilution. After washing the sections in PBS, they were incubated for 2 h at room temperature with the secondary rabbit anti-goat Alexa 568, goat anti-rabbit 568 and rabbit anti-mouse 568 (Invitrogen, Carlsbad, CA, USA) antibodies. SYTOX green nucleic acid stain (Invitrogen) diluted in PBS (1:500 dilution) was incubated for 10 min for nuclear counterstaining. Slides were mounted in a fluoromount (Sigma-Aldrich) medium for further analysis using a laser scanning confocal microscope (TCS SP5; Leica Microsystems GmbH, Heidelberg, Germany). Immunohistochemistry controls were performed by omission of the primary antibody in a sequential tissue section.

### 2.6. Statistical Analysis

Data were recorded in an Excel database and statistical analysis was performed using SPSS software version 20.0 (SPSS Inc., Chicago, IL, USA). The Kolmogorov–Smirnov test was used to assess sample distribution. Given the non-parametric distribution of most of the data, the differences between the EPI (episcleral) and Ms (microspheres) models were evaluated using the Mann–Whitney U test, and the longitudinal changes recorded in each eye over the 24-week study period were evaluated using a paired Wilcoxon test. All values were expressed as mean ± standard deviations. Values of *p* < 0.05 were considered to indicate statistical significance. To avoid a high false-positive rate, the Bonferroni correction for multiple comparisons was applied and the level of significance for each variable was established based on Bonferroni calculations.

## 3. Results

### 3.1. Ophthalmological Signs and IOP

No cases of infection, severe intraocular inflammation or retinal detachment were found in either model, but the corneal surface was better preserved in the Ms20/10 model, allowing visualization of the microspheres in the anterior chamber of the eye over the 24 weeks and producing higher OCT and ERG recordings (Figure 2). Cataract formation was not detected in the Ms20/10 model. However, reversible cataracts were observed in two animals in the EPIm, as described by [21]. Under 3 min were needed to take IOP recordings, as recommended by [25], and approximately 7 min were needed under the Ms20/10 model to perform each ocular injection (from induction to recovery); longer surgical times were required with the EPIm.

Under the Ms20/10 model, RE IOP progressively increased over the study, with levels exceeding 20 mmHg (OHT) from week 12 onwards. The LE also experienced a progressive increase and even reached OHT levels from week 19 onwards. RE IOP values were significantly higher (*p* < 0.05) than LE values. Under the EPIm, the RE reached OHT levels between weeks 1 and 10 but then steadily decreased. The contralateral LE also experienced an IOP increase, although it was always lower than the RE. Comparative analysis of the two models revealed that RE values under the EPIm were significantly higher (*p* < 0.05) than under the microsphere model in the first half of the study (Figure 3A).

The percentage of eyes with OHT was also analyzed in each model. REs in the Ms20/10 model showed a higher OHT percentage at halfway and at later exploration times; the highest percentage of eyes with OHT was reached at week 19 (83.3%), while the LE reached this value one month later. The RE presented higher percentages of OHT than the LE throughout the period. However, under the EPIm, the highest percentage (91.3%) was observed earlier (specifically, at week 8) before subsequently decreasing. On average, the EPIm showed a higher percentage of OHT eyes than the Ms20/10 model in both the right and left eyes (Figure 3B).

### 3.2. In Vivo Neuroretinal Analysis by Optical Coherence Tomography

Under the Ms20/10 model, the RE retina, RNFL and GCL showed a statistical tendency to decrease in thickness over time (Table 1). However, fluctuations were observed in both eyes, mainly in the retina and RNFL, although they also later occurred in the LE. RE vs. LE comparison showed significantly greater thickness in the RE, mainly in the retina, at week 8 (Appendix A (Appendix A)). Under the EPIm, both eyes also showed a statistically significant decrease in thickness over time in the retina, RNFL and, principally, GCL. The RE retina also showed an early increase in thickness (Figure 4).

Both models were compared and no statistical differences were found in thickness in most of the parameters analyzed. Statistical differences were observed in only 6 of the 135 parameters: 4 in the retina, none in the RNFL, and just 2 in the GCL. The EPIm showed a tendency towards retinal thickening and GCL thinning over time (Table 2). The percentage loss in thickness was also quantified: in both models, and at every point in time, the RE experienced the highest percentage thickness loss in the inner sectors of the superior–inferior axis in the retina and RNFL (Figure 5).

Figure 6 showed fluctuations but with a tendency to higher percentage loss over time. The highest average percentage loss in both eyes and models was in the GCL, followed by the RNFL and finally the retina, and an even higher percentage thickness loss was observed in the LE. On average, the EPIm showed a bigger loss than the Ms20/10 model.

The loss rate expressed in microns per mmHg and day extracted from the all-sector average was also quantified in both eyes and the models were compared. The highest loss rate in the OHT-induced eye (RE) was found in the RNFL followed by the GCL and finally the retina. Thus, based on IOP (mmHg), the RNFL was the structure affected earliest and most severely. On average, the EPI and Ms20/10 models showed exactly the same loss rate in the RNFL (−5 nm/mmHg/d), the LE showed a higher loss (−7 nm/mmHg/d) and a similar situation occurred with the GCL (Table 3).

### 3.3. In Vivo Electroretinogram

Both OHT models showed a progressive decrease in neuroretinal function at weeks 12 and 24 in dark- and light-adapted tests. In general, the EPIm presented statistically slower (in latency) or lower (in amplitude) recordings than the Ms20/10.

In dark-adapted cells under both models, the a-wave (from photoreceptors) showed a smaller amplitude but faster response than the b-wave (from bipolars and intermediate cells), and both also presented the lowest recordings within the initial phases of stimuli using the lightest flash intensities. The EPIm presented statistically slower records in the a-wave up to week 12 but also in the b-wave over the course of the study (Figure 7A,B), as well as a lower amplitude in the a-wave (Figure 7C).

A light-adapted test using the PhNR protocol was performed to specifically study RGC function. In this case, the EPIm showed a statistically slower response at week 12 that inverted later (Figure 7E), as also occurred with the scotopic ERG test. Furthermore, a lower RGC amplitude was found throughout (Figure 7F). Under the light-adapted test, the photoreceptor’s functionality (a-wave amplitude) in the EPIm statistically decreased compared to the Ms20/10 model at the end of the study.

### 3.4. Pathological Findings

To compare retinal morphological alterations between the Ms20/10 and EPI models, six rat eyes (3 LEs and 3 REs) for each glaucoma model were analyzed after 6 months of increased chronic IOP. A similar RGC loss was previously demonstrated in both models by using the Brn3a marker [20]. In this study, we performed a more exhaustive evaluation of the retina with different markers in order to analyze the aggressivity of each model.

Neuroglia was analyzed using anti-GFAP and anti-GS antibodies, markers of astrocytes [26] and Müller cells [27], respectively. As expected, astrocytes were confined to the retinal ganglion cell layer in contralateral control eyes (Figure 8). In hypertensive eyes, GFAP expression was observed in Müller cell processes (gliosis) under both the Ms20/10 and EPI models (Figure 8). This is consistent with previous papers indicating that glial activation is present during glaucoma [28]. The retinal topography and morphology of Müller cells were similar when comparing control and hypertensive eyes, as well as when comparing the two glaucomatous models (Figure 9). Müller cell bodies were localized in the inner nuclear layer and Müller cell processes formed the internal and external limiting membranes (Figure 9). However, the expression of glutamine synthetase (GS) was diminished in EPI Müller cells, both in contralateral control and hypertensive eyes, the decrease being much more evident in glaucomatous eyes (Figure 9). This is in accordance with previous studies of the retinas of dogs [29] and DBA/2J mice [30] that show diminished GS expression in Müller cells during glaucoma. GS is a major enzyme involved in the metabolism of glutamate in Müller cells. It catalyzes the conversion of glutamate to glutamine, which is an essential part of the cycling of glutamate between neurons and glia [31]. Excess extracellular glutamate not taken up by Müller cells is toxic to neurons and is likely to potentiate retinal cell loss in glaucoma [30].

Microglia was evaluated using the ionized calcium-binding adaptor molecule 1 (Iba1) marker (Figure 10). Iba1 is a 17-kDa protein whose expression is restricted to macrophages, including microglia [32]. In control eyes and Ms20/10 hypertensive eyes, microglial cells were hardly visible in the retinal histological sections, though isolated microglial cells were occasionally observed (Figure 10). In contrast, in EPI hypertensive eyes it was very common to find groups of three or four activated microglial cells by microscopic field (Figure 10).

Next, bipolar cells and cones were analyzed in the Ms20/10 and EPI glaucomatous models. Immunostaining with the anti-PKCα antibody, a specific marker of rod bipolar cells [33], was performed. PKCα is a calcium-dependent protein kinase that has been implicated in ganglion cell loss during glaucoma [34]. PKCα levels in dendrites of rod bipolar cells remarkably decreased after acute high IOP [34]. However, such a decrease of PKCα was not observed in rod bipolar cell dendrites in either the Ms20/10 or EPI models (Figure 11). In contrast, a morphological alteration, compatible with the dilation of axon terminals, was observed only in the rod bipolar cells in EPI glaucomatous eyes (Figure 12).

Cones were evaluated using the anti-cone arrestin antibody (Figure 13). No remarkable differences were observed between control and hypertensive eyes, nor when the two glaucoma models were compared (Figure 13).

Our results suggested that the Ms20/10 model is a less aggressive glaucoma model because it does not present the alterations found in the EPI model, which presented diminished GS expression, an increase in microglial cells and dilation of axon terminals in rod bipolar cells.

## 4. Discussion

Intraocular pressure increase is a major risk factor associated with the onset and progression of glaucoma. Development of animal models of glaucoma simulating the conditions present in humans is a challenge as progressive RGC death and glaucomatous optic disc neuropathy have to be present.

The method of injecting microparticles into the anterior segment to produce ocular hypertension is not new, in fact the intracameral injection of polystyrene or paramagnetic microbeads are among the most used methods to create artificial glaucomatous lesions [35]. The approach presented in this work takes advantage of these aforementioned models, since it produces physical blockage of the trabecular meshwork, but also has some additional advantages. On the one hand, the biodegradable polymer employed to fabricate the microparticles is biocompatible, which keeps ocular structures healthier and free of inflammatory synechiae but also avoiding the application of electromagnetic fields [36]. In addition, PLGA microparticles presented an additional advantage as they are excellent systems for the controlled release of active substances, both low molecular weight and bio-engineered compounds [37] for long periods, some of which could be potentially used to produce additional damage in the anterior segment and thus create new models of ocular hypertension. In a previous preliminary study [20], our group demonstrated that intracameral injection of different-sized fractions of biodegradable PLGA Ms can be used to produce a progressive increase in IOP, with the subsequent structural degeneration of the neuroretina. The changes were detected by OCT and by lower RGC counts on histological examinations over 8 weeks of follow-up. In that preliminary study, it was established that the glaucomatous damage provoked by PLGA Ms injection was comparable to that of the well-established Morrison model (EPIm in our study), with the best results being achieved with the Ms 10–20 µm fraction size. The 10–20 µm particle size fraction was selected as the optimal size range to promote a progressive and sustained increase in IOP in rat eyes with minimal ocular surface alteration [20]. After repeated intracameral injection, Ms at different stages of degradation were the cause of progressive obstruction of the trabecular meshwork, promoting aqueous humor accumulation and, hence, a progressive increase in IOP. Furthermore, this accumulation was also associated with macrophage activation in the area. The presence of such inflammatory cells can produce an acidic environment in the extracellular fluids [38], which is an additional advantage, as an acidic environment is also present in glaucoma.

However, as previously mentioned, most glaucoma animal model evaluation studies are conducted with short to medium follow-up times. This is far from the real-world conditions of glaucoma in humans, in which it is typically a chronic pathology characterized by slow and progressive neurodegeneration of the retina and the optic nerve. The current animal models of glaucoma, in which artificial generation of acute retinal damage occurs rapidly, are unsuitable for in-depth study of the physiopathogenic milestones of chronic glaucoma, or for evaluation of the therapeutic effect of antiglaucomatous therapies. This shortcoming becomes even more evident when these therapies are formulated in sustained drug-delivery systems.

In this paper, the study period has been prolonged to 24 weeks, demonstrating that the animal model proposed (named Ms20/10) is not only able to generate retinal neurodegeneration comparable to that of Morrison’s model but also presents several important advantages: (1) the new modified model produces a hypertensive curve with a progressive and moderate increase in IOP over 24 weeks, more similar to conditions found in open-angle primary glaucoma, and with fewer fluctuations and therefore less variability than EPIm; (2) the Ms20/10 model produces milder glaucomatous degeneration, since the histological examinations do not detect features of aggression such as severe glial dysfunction or morphological cell alterations; (3) the Ms20/10 model requires fewer inductive reinjections than the EPI model [39], thus decreasing action on the animal, improving its welfare and reducing the associated costs; (4) considering the nature of the microspheres employed, as mentioned before, the particles could be loaded with substances able to damage the trabecular meshwork that could be slowly released in the intracameral environment and further reduce the number of reinjections required to develop the disease; (5) the injected biocompatible material (PLGA) maintains healthy ocular surface throughout the 24 weeks of study, which makes it possible to carry out serial in vivo explorations with effective OCT and ERG; and (6) it allows for adequate space–time analysis of glaucomatous pathology.

To evaluate neuroretinal changes over prolonged study periods and reduce the number of animals used, serial in vivo explorations are ideal. The ERG and OCT tests have demonstrated their efficacy in evaluating retinal degeneration and the effect of therapies [40,41]. In our study, functional (ERG) and structural (OCT) tests were performed sequentially over 24 weeks and postmortem examinations were conducted at the end of the study.

Porciatti et al. [40,41] reported dysfunction of the outer layers, RGC and axons and dendrites, which precedes cell death. According to our results, this dysfunction varies depending on the degree of IOP elevation.

Our study showed that in the Ms20/10 model, the viability of the cones was preserved throughout follow-up by maintaining the ERG amplitude after high-intensity stimulus, which was corroborated in histology by not finding remarkable changes (Figure 7C and Figure 13). The EPI model (with higher, earlier and more aggressive levels of IOP) generated poorer signals in scotopic ERG—measured as increased latency in the response of rods and bipolars (after mild stimulation) (week 12) (Figure 7A,B), in agreement with the dilation of axonal terminals of rod-bipolar cells observed in histology (week 24)—as well as lower amplitude, although a smaller number of photoreceptor and intermediate cells was not detected. With the photopic test, the EPIm recorded lower ERG responses, suggesting a loss of cone-RGC function with anatomical-structural preservation of photoreceptors (Figure 7F and Figure 13). These results corroborate early dysfunction detected by ERG prior to clear evidence of structural loss or damage.

In addition, the Ms20/10 model recorded higher PhNR latencies coinciding with higher IOP levels (24 weeks). This same milestone was detected in the EPI model, which recorded an increase in PhNR latency at week 12, and latency shortened at later times (week 24) when IOP levels were lower. Our prolonged study period allowed us to observe a window (in our case 12 weeks) in which live RGCs recover functionality—a threshold time in which application of potential neuroprotective therapy could be effective. On the other hand, in week 24 the EPI model recorded lower PhNR amplitudes. This implies a lower number of functioning RGCs, although it is possible that those that did work overexerted themselves to compensate. Although ganglion dysfunction was also observed in the Ms20/10 model, this occurred later and showed less involvement of the external layers, demonstrating that it is a less aggressive model.

Histological results corroborated the ERG findings. However, although the ERG test allows functional monitoring in vivo, it has several handicaps. On the one hand, the ERG determination can be affected by optical media opacity. In this regard, the ocular surface in the Ms20/10 model was better preserved than EPIm, which in part, the poorer signal recorded could be due to this disadvantage and produce more variable results. The Ms20/10 model, unlike the EPIm and other animal models of glaucoma induced by intracameral injection of non-biodegradable Ms, avoids this potential variability and ensures that signal loss is secondary to neurodegeneration by maintaining healthy anterior ocular structures. On the other hand, ERG results can vary due to factors such as anesthetic depth, dark-adaptation time, or even pupil size [42]; and it is also a lengthy test with the consequent prolongation of the anesthetic time and risk of unexpected death.

In contrast, OCT has the advantages of being a simple, fast and objective test that has demonstrated an adequate histological correlation [43]. Our findings are in agreement with those of previous authors [15,19,40], who observed that the sectors of the upper-inferior axis of the RNFL were the most affected (mainly the S sector) after IOP elevation. Physiologically, rats present more fiber thickness in the sectors of the vertical axis [44], whose axonal viability could be compromised in case of noxa. Furthermore, at later stages, greater involvement of the nasal sector in the GCL was detected, which coincides with the horizontal axis (nasal-temporal) where healthy rats present greater RGC density [45]. Finally, greater retinal loss was detected in internal sectors up to 24 weeks’ follow-up. This contrasts with previous studies [20,46,47], which showed greater thinning in external sectors at early stages. These discrepancies could be due to early thinning influenced by the growth of the eye at the expense of peripheral areas, as well as to immune infiltration from peripheral choroids that underestimates the external loss described in glaucoma [48].

In order to compare neuroretinal loss objectively, it was normalized according to the thickness loss measured by OCT per day and per mmHg of IOP rise, as previously done by other authors [20,40] up to week 8, finding similar rates. Our model permitted, for the first time, a much longer normalized study, conducted over 24 weeks, of the RNFL and, in a novel way, of the GCL, thanks to automated segmentation. This normalization allowed us to identify the RNFL as the earliest affected parameter as a function of IOP, and corroborates that axons are the structure first affected by OHT in the optic nerve head [49,50]. However, the highest percentage loss was observed in the GCL, which could be due to the decline of the synaptic tree prior to cell death, detected by ERG [51,52].

When quantifying IOP as an independent variable, it was found that each increase in mmHg always caused the same damage, since both models, even at later stages, presented a similar loss rate in the RNFL and a very similar one in the GCL. In other words, the greater the OHT (increased mmHg) the greater the damage. Therefore, this suggests that the differences in the progression or severity of the final neurodegenerative damage are a consequence of the extent of OHT, as well as of the inductive model per se when triggering glial, immune [48] and proinflammatory reactivity in different ways, reflected as changes occurring in the entire retina.

The main differences between models were observed in the retina protocol (Table 2 and Table 3). The EPIm showed a lower early loss, possibly due to underestimation of neurodegeneration (considered a decrease in thickness as measured by OCT), to an increase in cellular soma prior to death [53] and to higher glial reactivity and inflammatory infiltration (Figure 8 and Figure 10) coinciding with the increase in IOP. However, these differences between models disappeared at later stages, when the Ms20/10 model reached higher levels of IOP.

Interestingly, a neurodegenerative pattern was detected in both models, comprising fluctuations in the contralateral eye (similar to the induced eye) but one month later. In this respect, recent studies (albeit all of them involving a maximum of 6 weeks of follow-up and more aggressive OHT-inducing models) have shown bilateral detection of inflammatory mediators and immune activation in the non-induced counterpart eye [47,54]. The EPI (more hypertensive) model showed mild astrogliosis in the opposite eye, which corroborates the findings of previous authors and the unsuitability of using the opposite eye as a control in future studies [47,55]. However, a small increase in IOP could trigger a T-cell-mediated neurodegenerative process [56]. To our knowledge, this study is the first to demonstrate contralateral neurodegeneration in a long-term (24 weeks’ follow-up) chronic glaucoma model with a smooth and progressive IOP increase that simulates primary open-angle glaucoma. This recent finding would suggest that when faced with diagnosed unilateral glaucoma, it could be advisable to treat the contralateral eye early. It is known that the secondary neurodegenerative process advances, even with adequate pressure control, and therefore possible neurodegenerative dissemination could have started in the rest of the visual pathway [57].

The underestimation of OCT-measured neurodegeneration found between models was also observed between eyes. In the injected right eyes, less neuroretinal loss was measured—coinciding with greater glial response—than in the corresponding contralateral eyes (Figure 6 and Table 3). Previous authors have shown an adequate OCT/histology correlation [43]. However, our study suggests that OCT measurements should be evaluated individually according to the stage of glaucoma in the patient, since greater damage can be under-detected if neurodegeneration is considered only as a loss of neuroretinal thickness [44].

Very little is known about the effects on glial responses of different states or temporal sequences during the course of glaucoma neurodegeneration. Glial activation may be sufficient to develop an adaptive para-inflammatory process to restore homeostasis, but its chronicity leads to cell death [58]. Our results show a more robust involvement of neuroglia (suggested as the major drivers of neuroinflammatory damage in the internal retina [59]) than microglia. Alteration of intraretinal microglia is selectively detectable in eyes that progress to glaucoma in later stages, and early microgliosis has a strong correlation with the severity of optic nerve degeneration [60], as observed in the EPI model. Recently, it was demonstrated in the brain that astrocytes and microglia play orchestrated roles in the elimination of cellular debris and that in pathological conditions astrocytes are capable of generating compensatory plasticity [61]. The EPI glaucoma model presented more astrocytic alteration, possibly in a compensatory way, but overall was less effective.

## 5. Conclusions

For all the aforementioned reasons, even though both models are comparable, the Ms20/10 model simulates the hypertensive curve and temporal neurodegenerative sequence of chronic glaucoma better, being smoother and less aggressive and allowing longer exploratory times (up to 24 weeks). To our knowledge, it would be the most suitable model with which to study the progression of glaucoma, as well as with which to evaluate possible neuroprotective/antiglaucomatous therapies, especially those involving intraocular drug-delivery systems.

## Figures and Tables

**Figure 1 biomedicines-09-00682-f001:**
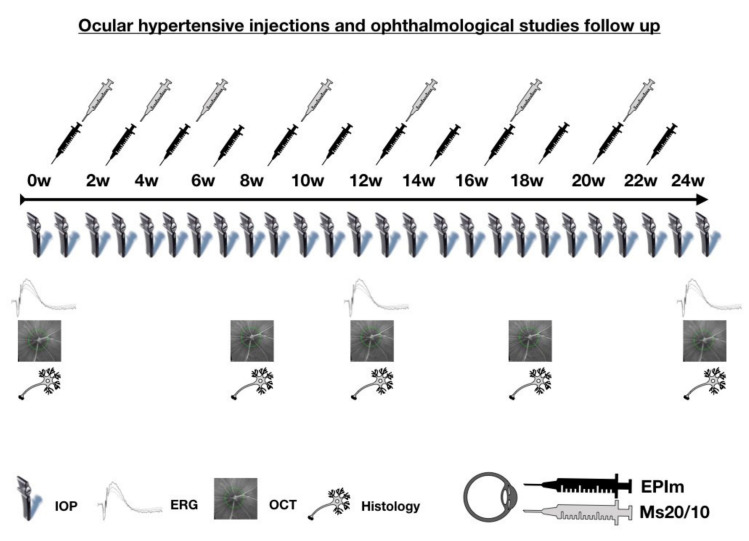
Ocular hypertensive injections and ophthalmological studies. Abbreviations: Ms20/10: microsphere model; EPIm: episcleral sclerosis model; IOP: intraocular pressure; OCT: optical coherence tomography; ERG: electroretinography; w: week.

**Figure 2 biomedicines-09-00682-f002:**
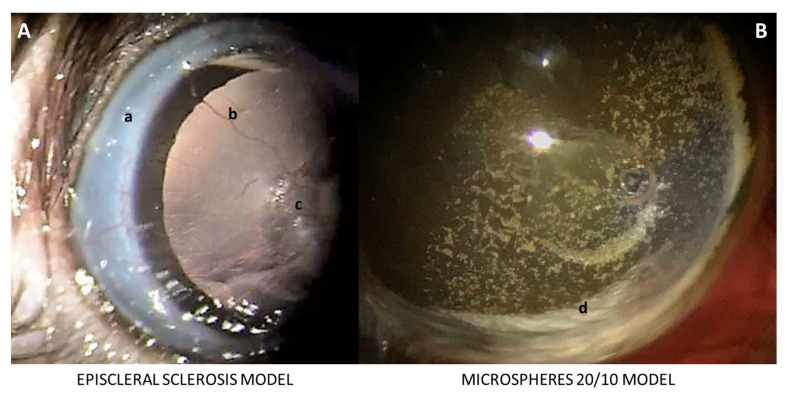
Ocular surface of rat eye in ocular hypertensive models. (**A**) Episcleral sclerosis model. a: sclerosis of episcleral veins; b: corneal neovascularization; c: corneal leucoma. (**B**) Microsphere 20/10 model. d: microspheres of poly lactic-co-glycolic acid (PLGA) in the anterior chamber of the rat eye.

**Figure 3 biomedicines-09-00682-f003:**
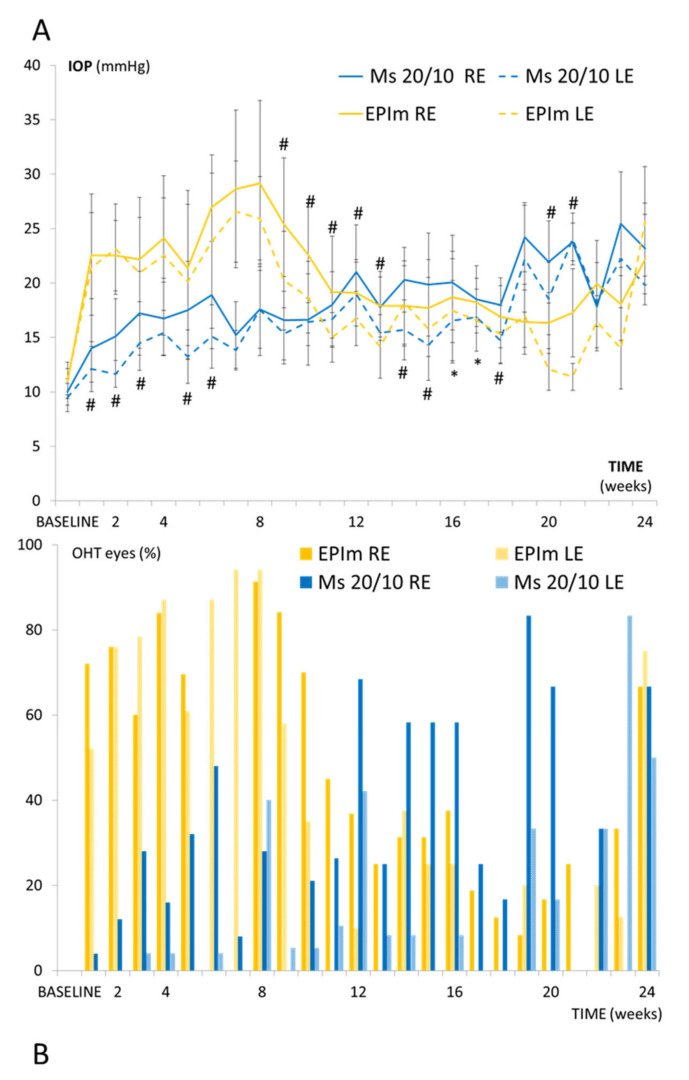
(**A**): Intraocular pressure comparison over 6 months in both ocular hypertensive models. (**B**): Percentage of ocular hypertensive eyes (>20 mmHg) in the EPIm vs. Ms20/10 models over 6 months’ follow-up. Abbreviations: Ms20/10: microsphere model; EPIm: episcleral sclerosis model; RE: right eye; LE: left eye; IOP: intraocular pressure; OHT: ocular hypertension; %: percentage; *: statistical significance *p* < 0.05; #: statistical significance *p* < 0.02, for Bonferroni correction for multiple comparisons.

**Figure 4 biomedicines-09-00682-f004:**
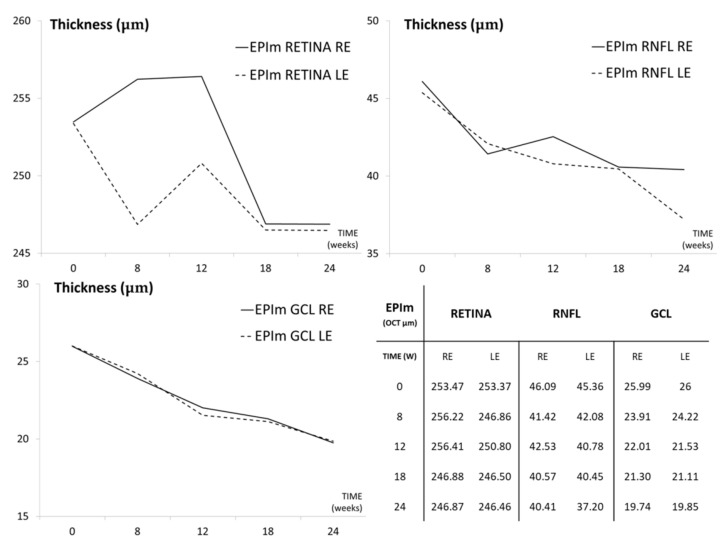
Structural analysis of neuroretina of both eyes by optical coherence tomography under the episcleral model. Abbreviations: EPIm: episcleral sclerosis model; RE: right eye; LE: left eye; OCT: optical coherence tomography; RNFL: retinal nerve fiber layer; GCL: ganglion cell layer; average thickness in microns (μm).

**Figure 5 biomedicines-09-00682-f005:**
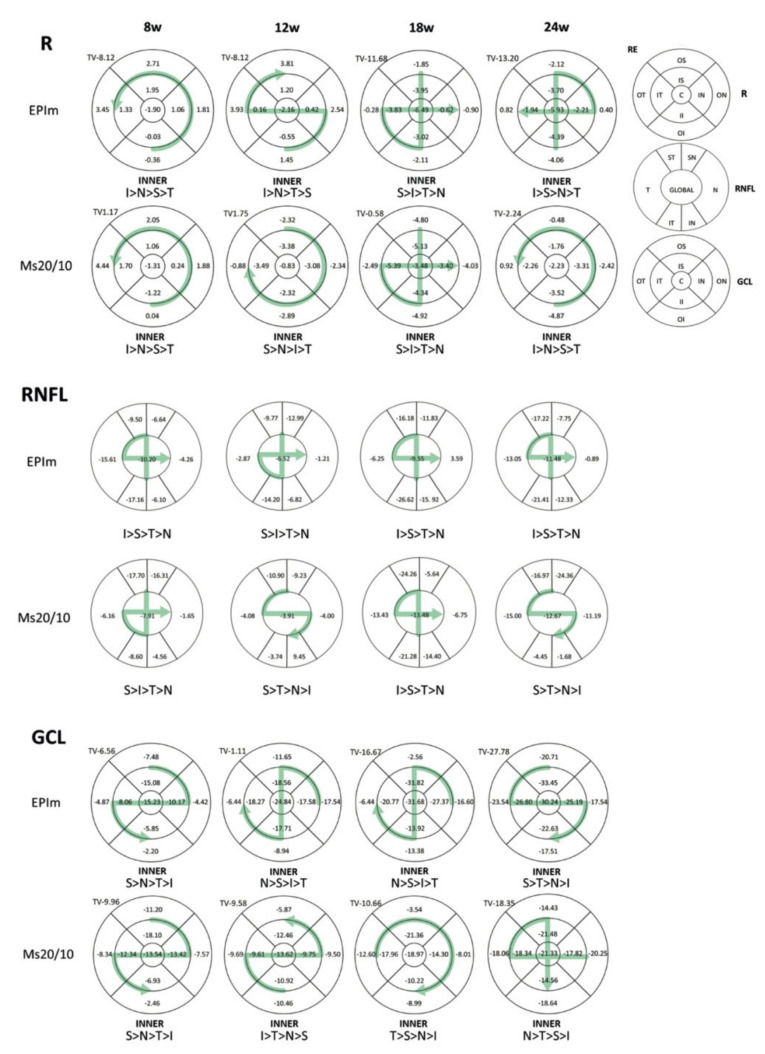
Neuroretinal percentage loss by optical coherence tomography sector and loss trend in both ocular hypertensive models. Abbreviations: EPIm: episcleral sclerosis model; Ms20/10: microsphere model; RE: right eye; R: retina; C: central; II: inner inferior; OI: outer inferior; IS: inner superior; OS: outer superior; IN: inner nasal; ON: outer nasal; IT: inner temporal; OT: outer temporal; RNFL: retinal nerve fiber layer; IT: inferior temporal; IN: inferior nasal; ST: superior temporal; SN: superior nasal; N: nasal; T: temporal; GCL: ganglion cell layer; TV: total volume; I: inferior; S: superior; N: nasal; T: temporal; >: higher loss than.

**Figure 6 biomedicines-09-00682-f006:**
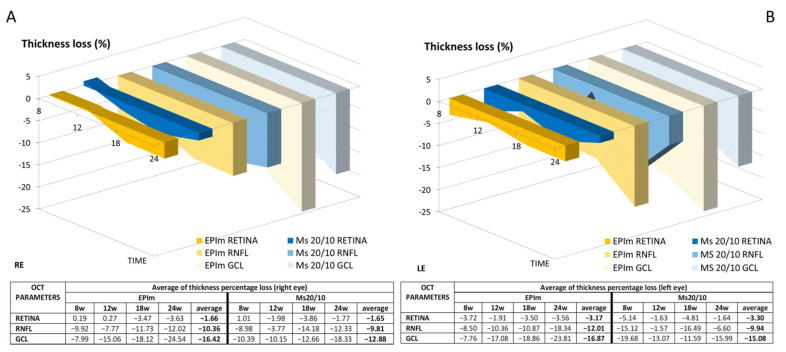
Percentage of thickness loss measured by OCT in both OHT models over 6 months’ follow-up. (**A**): Right eye (RE) analysis. (**B**): Left eye (LE) analysis. Abbreviations: EPIm: episcleral sclerosis model; Ms20/10: microsphere 20/10 model; RNFL: retinal nerve fiber layer; GCL: ganglion cell layer; w: week; %: percentage.

**Figure 7 biomedicines-09-00682-f007:**
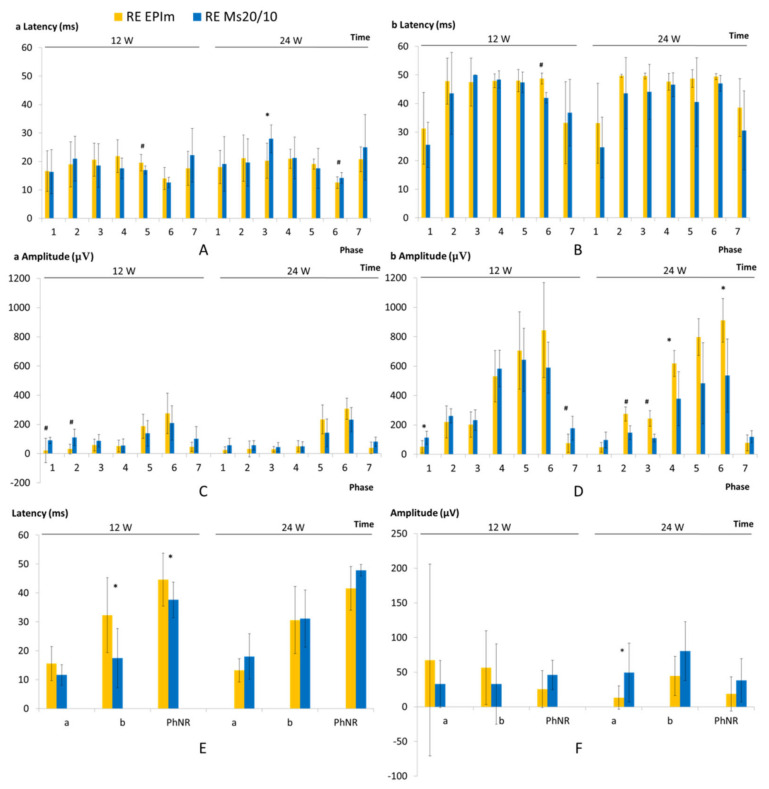
Comparison of functional neuroretinal measurements by electroretinogram between both ocular hypertensive models. Abbreviations: EPIm: episcleral sclerosis model; Ms 20/10: biodegradable microsphere 20/10 model. Scotopic electroretinogram: ERG. (**A**): latency of photorreceptors; (**B**): latency of intermediate cells; (**C**): amplitude of photoreceptors; (**D**): amplitude of intermediate cells; Rod response: phase 1. 0.0003 cds/m^2^, 0.2 Hz/s; phase 2: 0.003 cds/m^2^, 0.125 Hz/s; phase 3: 0.03 cds/m^2^, 8.929 Hz/s; phase 4: 0.03 cds/m^2^, 0.111 Hz/s; phase 5: 0.3 cds/m^2^, 0.077 Hz/s. Mixed rod-cone response: phase 6: 3.0 cds/m^2^, 0.067 Hz/s. Oscillatory potentials: phase 7: 3.0 cds/m^2^, 29.412 Hz/s explored by dark adaptation. PhNR: photopic negative response; (**E**): latency; (**F**): amplitude. a: a-wave expresses the functionality of photoreceptors; b: b-wave expresses the functionality of intermediate cells; PhNR-wave expresses the functionality of retinal ganglion cells. PhNR explored by blue light adaptation to a blue background (470 nm, 25 cds/m^2^), and a red LED flash (625 nm, −10 dB, 0.30 cds/m^2^, 1.199 Hz) as stimuli; w: week; ms: milliseconds; μV: microvolts; *: statistical significance *p* < 0.05; #: statistical significance *p* < 0.02, for Bonferroni correction for multiple comparisons.

**Figure 8 biomedicines-09-00682-f008:**
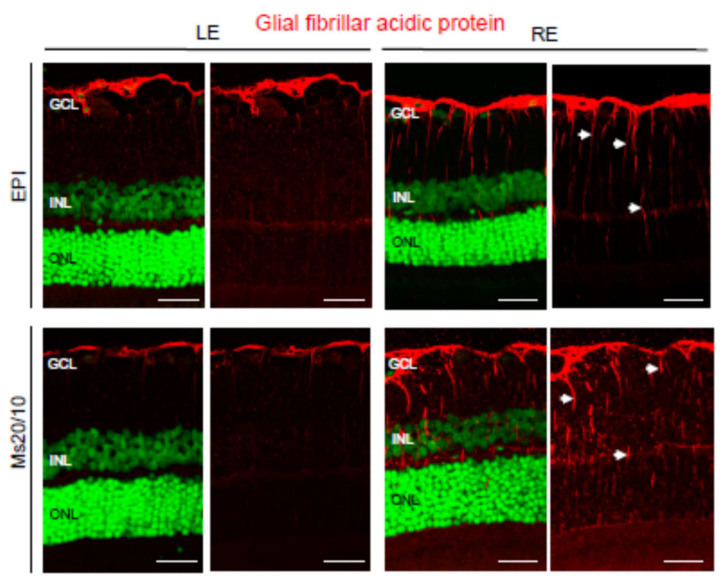
Glial fibrillary acidic protein (GFAP) expression in the glaucomatous EPI (episcleral) and Ms20/10 (microspheres) models. Representative images showed the location and total volume of GFAP observed in the retinas. Note the expression of GFAP in Müller cell processes (arrows) in hypertensive eyes. GCL: ganglion cell layer; INL: inner nuclear layer; ONL: outer nuclear layer; LE: left eye (control); RE: right eye (treated). Scale bar: 30.49 μm.

**Figure 9 biomedicines-09-00682-f009:**
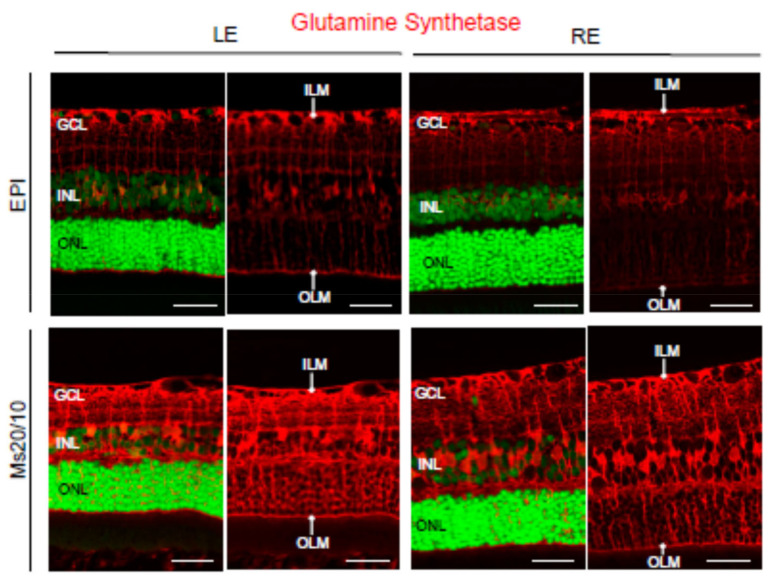
Glutamine synthetase (GS) expression in the glaucomatous EPI and Ms20/10 models. Representative images showed the location and total volume of GS observed in the retinas. GCL: ganglion cell layer; INL: inner nuclear layer; ONL: outer nuclear layer; LE: left eye (control); RE: right eye (treated); ILM: inner limiting membrane; OLM: outer limiting membrane. Scale bar: 34.24 μm.

**Figure 10 biomedicines-09-00682-f010:**
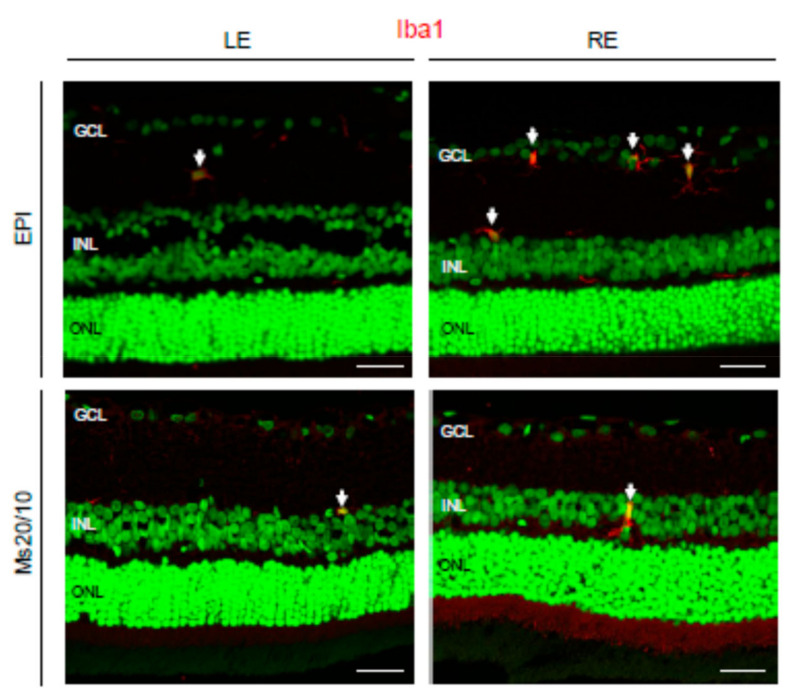
Ionized calcium-binding adaptor molecule 1 (Iba1) expression in the glaucomatous EPI and Ms20/10 models. Representative images showed the location of microglial cells (arrows) marked by Iba1 in the retinas. GCL: ganglion cell layer; INL: inner nuclear layer; ONL: outer nuclear layer; LE: left eye (control); RE: right eye (treated). Scale bar: 31.25 μm.

**Figure 11 biomedicines-09-00682-f011:**
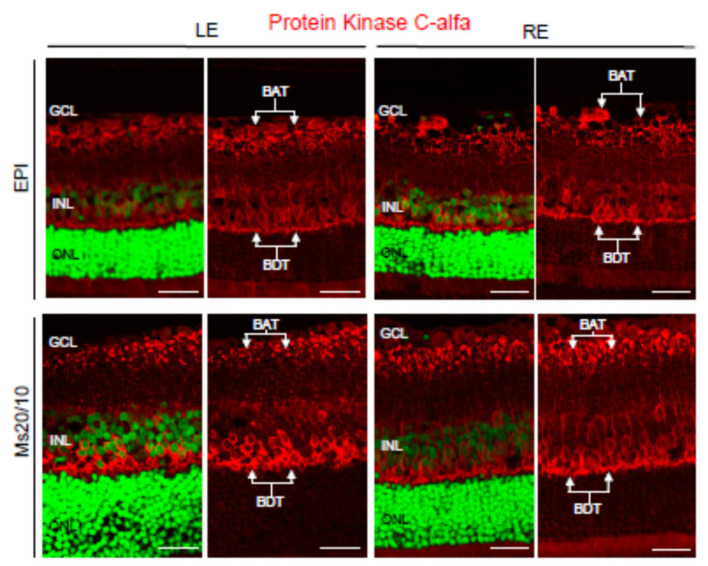
Calcium-dependent protein kinase C alfa (PKCα) expression in the glaucomatous EPI and Ms20/10 models. Representative images showed the location and total volume of PKCα observed in the retinas. GCL: ganglion cell layer; INL: inner nuclear layer; ONL: outer nuclear layer; LE: left eye (control); RE: right eye (treated); BAT: bipolar axon terminals; BDT: bipolar dendrite terminals. Scale bar: 34.24 μm.

**Figure 12 biomedicines-09-00682-f012:**
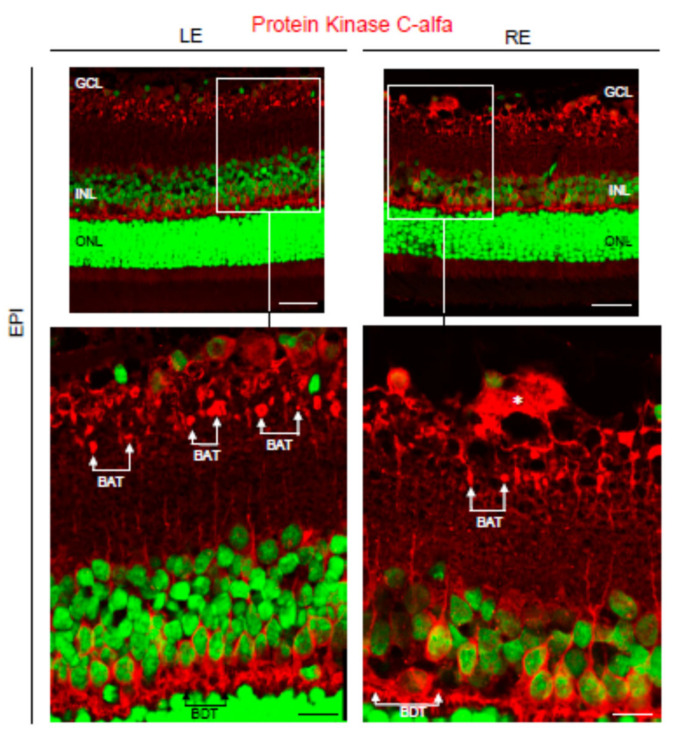
Calcium-dependent protein kinase C alfa (PKCα) expression in the glaucomatous EPI model. Note the dilation (*) of axon terminals in rod bipolar cells under the EPI model. GCL: ganglion cell layer; INL: inner nuclear layer; ONL: outer nuclear layer; LE: left eye (control); RE: right eye (treated); BAT: bipolar axon terminals; BDT: bipolar dendrite terminals. Scale bar: 35.71 μm and scale bar (inset): 13.37 μm.

**Figure 13 biomedicines-09-00682-f013:**
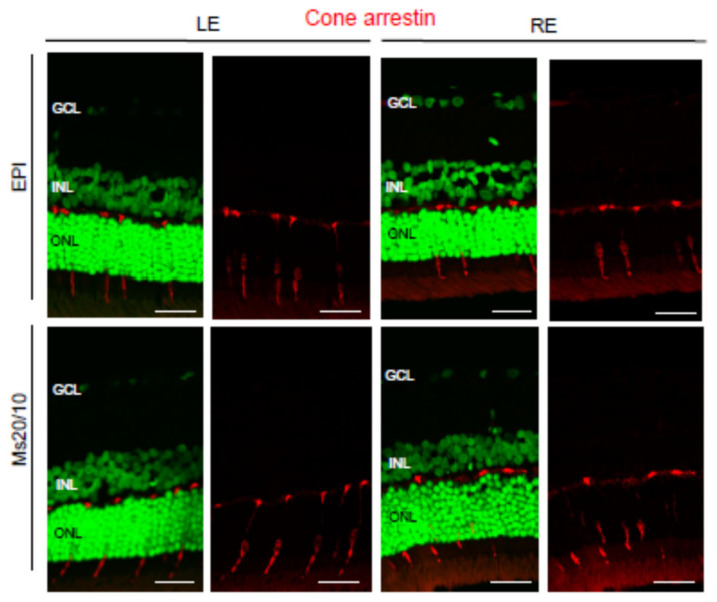
Cone arrestin expression in glaucomatous the EPI and Ms20/10 models. Representative images showed the location and total volume of cone arrestin observed in the retinas. GCL: ganglion cell layer; INL: inner nuclear layer; ONL: outer nuclear layer; LE: left eye (control); RE: right eye (treated). Scale bar: 30.86 μm.

**Table 1 biomedicines-09-00682-t001:** Structural analysis of right-eye neuroretina by optical coherence tomography (OCT) (Ms20/10 model).

Right Eye, Microsphere 20/10 Model													
OCT Parameters (μm)	Baseline	8 w			12 w			18 w			24 w		
	Mean ± SD	Mean ± SD	%Ch	*p*	Mean ± SD	%Ch	*p*	Mean ± SD	%Ch	*p*	Mean ± SD	%Ch	*p*
**Retinal thickness**	↑			↓			↓			↑			
central	271.04 ± 13.48	267.50 ± 10.68	−1.31	0.347	268.80 ± 17.72	−0.83	0.5	261.60 ± 9.91	−3.48	0.893	265.00 ± 22.45	−2.23	0.6
inner inferior	256.54 ± 7.39	253.42 ± 8.59	−1.22	0.237	250.60 ± 8.50	−2.32	0.221	245.40 ± 10.13	−4.34	0.043	247.50 ± 14.48	−3.52	0.116
outer inferior	246.33 ± 5.94	246.42 ± 10.71	0.04	0.802	239.20 ± 6.09	−2.89	0.345	234.20 ± 2.58	−4.92	**0.042**	234.33 ± 10.25	−4.87	**0.028**
inner superior	251.08 ± 7.70	253.75 ± 25.36	1.06	0.846	242.60 ± 15.50	−3.38	0.345	238.20 ± 6.68	−5.13	**0.041**	246.67 ± 16.23	−1.76	0.752
outer superior	250.21 ± 6.37	255.33 ± 20.02	2.05	0.573	244.40 ± 7.43	−2.32	0.581	238.20 ± 6.18	−4.80	**0.043**	249.00 ± 10.67	−0.48	0.753
inner nasal	253.21 ± 6.93	253.83 ± 15.81	0.24	0.931	245.40 ± 9.39	−3.08	0.144	244.60 ± 7.63	−3.40	**0.043**	244.83 ± 15.80	−3.31	**0.027**
outer nasal	248.00 ± 6.03	252.67 ± 16.90	1.88	0.608	242.20 ± 9.09	−2.34	0.683	238.00 ± 5.61	−4.03	**0.043**	242.00 ± 10.90	−2.42	0.116
inner temporal	253.04 ± 10.20	257.33 ± 28.91	1.7	0.928	244.20 ± 8.10	−3.49	0.345	239.40 ± 5.45	−5.39	**0.043**	247.33 ± 14.41	−2.26	0.078
outer temporal	248.38 ± 7.42	259.42 ± 26.63	4.44	0.365	246.20 ± 10.08	−0.88	0.225	242.20 ± 2.86	−2.49	0.078	250.67 ± 12.43	0.92	0.5
total volume	1.71 ± 0.36	1.73 ± 0.05	**1.17**	**0.038**	1.74 ± 0.06	1.75	0.126	1.71 ± 0.36	−0.58	0.5	1.75 ± 0.08	2.24	**0.027**
**RNFL thickness**	↓			↑			↓			↑			
global	46.00 ± 4.40	42.36 ± 6.26	−7.91	**0.04**	44.20 ± 3.27	−3.91	0.345	39.80 ± 2.77	−13.48	**0.042**	40.17 ± 8.65	−12.67	**0.043**
inferior temporal	46.75 ± 6.52	42.73 ± 7.86	−8.60	**0.008 ^#^**	45.00 ± 6.96	−3.74	0.99	36.80 ± 4.65	−21.28	**0.042**	44.67 ± 12.12	−4.45	0.207
inferior nasal	46.96 ± 6.19	44.82 ± 5.87	−4.56	0.814	51.40 ± 6.80	9.45	0.99	40.20 ± 5.58	−14.40	0.068	46.17 ± 10.77	−1.68	0.458
superior temporal	49.38 ± 8.15	40.64 ± 11.73	−17.70	**0.012 ^#^**	44.00 ± 9.43	−10.90	0.08	37.40 ± 4.09	−24.26	**0.043**	41.00 ± 12.61	−16.97	**0.046**
superior nasal	39.00 ± 8.20	32.64 ± 16.72	−16.31	0.548	35.40 ± 4.03	−9.23	**0.042**	36.80 ± 7.53	−5.64	**0.043**	29.50 ± 9.99	−24.36	**0.046**
nasal	43.54 ± 6.15	42.82 ± 6.33	−1.65	0.878	41.80 ± 6.90	−4.00	0.144	40.60 ± 5.94	−6.75	**0.042**	38.67 ± 12.06	−11.19	0.075
temporal	49.21 ± 8.21	46.18 ± 9.71	−6.16	0.095	47.20 ± 6.09	−4.08	0.683	42.60 ± 5.89	−13.43	**0.042**	41.83 ± 8.32	−15.00	**0.027**
**GCL thickness**	↓			≈↓			≈↓			↓			
central	22.46 ± 2.10	19.42 ± 3.23	−13.54	**0.033**	19.40 ± 2.70	−13.62	0.066	18.20 ± 2.49	−18.97	**0.043**	17.67 ± 3.55	−21.33	**0.027**
inner inferior	28.29 ± 1.57	26.33 ± 2.10	−6.93	**0.002 ^#^**	25.20 ± 1.64	−10.92	0.131	25.40 ± 1.51	−10.22	**0.042**	24.17 ± 3.25	−14.56	**0.042**
outer inferior	27.25 ± 1.35	26.58 ± 3.50	−2.46	0.441	24.40 ± 1.67	−10.46	0.066	24.80 ± 0.83	−8.99	**0.042**	22.17 ± 4.79	−18.64	**0.027**
inner superior	26.96 ± 1.92	22.08 ± 3.82	−18.10	**0.006 ^#^**	23.60 ± 3.64	−12.46	0.042	21.20 ± 1.09	−21.36	**0.043**	21.17 ± 3.97	−21.48	**0.042**
outer superior	25.71 ± 2.57	22.83 ± 3.71	−11.20	**0.037**	24.20 ± 2.77	−5.87	0.285	24.80 ± 0.83	−3.54	0.492	22.00 ± 5.21	−14.43	0.168
inner nasal	26.38 ± 2.33	22.83 ± 2.03	−13.42	**0.004 ^#^**	23.80 ± 2.38	−9.75	0.257	22.60 ± 1.81	−14.30	**0.042**	21.67 ± 5.78	−17.82	**0.027**
outer nasal	26.96 ± 1.75	24.92 ± 2.02	−7.57	**0.013 ^#^**	24.40 ± 2.07	−9.50	0.059	24.80 ± 0.83	−8.01	0.066	21.50 ± 3.93	−20.25	**0.027**
inner temporal	26.33 ± 2.18	23.08 ± 4.83	−12.34	**0.044**	23.80 ± 1.92	−9.61	0.063	21.60 ± 4.61	−17.96	**0.039**	21.50 ± 4.18	−18.34	0.058
outer temporal	27.46 ± 1.56	25.17 ± 3.38	−8.34	0.061	24.80 ± 2.86	−9.69	0.066	24.00 ± 2.82	−12.60	**0.042**	22.50 ± 3.14	−18.06	**0.026**
total volume	0.19 ± 0.01	0.17 ± 0.01	−9.96	**0.047**	0.17 ± 0.01	−9.58	0.059	0.17 ± 0.01	−10.66	**0.042**	0.15 ± 0.03	−18.35	**0.042**

Abbreviations: Ch, change; %Ch, percentage of change in thickness loss (with respect to baseline); GCL, ganglion cell layer; OCT, optical coherence tomography; RNFL, retinal nerve fiber layer; thickness in microns (μm); SD, standard deviation; w, weeks; *p* < 0.05 statistical significance (in bold); *p* < 0.02 ^#^ statistical significance with Bonferroni correction for multiple comparisons. Black upward arrow and white downward: fluctuation pattern.

**Table 2 biomedicines-09-00682-t002:** Right-eye structural neuroretinal measurements according to the ocular hypertensive models (EPIm vs. Ms20/10).

Re Structural Neuroretinal Measurements According to the Oht Model (EPIm vs. Ms20/10)										
OCT Parameters (μm)	Baseline		8 w		12 w		18 w		24 w	
	Mean ± SD	*p*	Mean ± SD	*p*	Mean ± SD	*p*	Mean ± SD	*p*	Mean ± SD	*p*
**Retinal Thickness**										
central	271.08 ± 14.04	0.904	265.92 ± 14.79	0.794	265.22 ± 18.72	0.655	253.50 ± 11.67	0.268	255.00 ± 18.17	0.253
	271.04 ± 13.48		267.50 ± 10.68		268.80 ± 17.72		261.60 ± 9.91		265.00 ± 22.45	
inner inferior	256.24 ± 0.14	0.904	256.17 ± 9.18	0.385	254.83 ± 9.19	0.478	248.50 ± 10.66	0.902	245.00 ± 12.91	0.668
	256.54 ± 7.39		253.42 ± 8.59		250.60 ± 8.50		245.40 ± 10.13		247.50 ± 14.48	
outer inferior	247.48 ± 6.75	0.520	246.58 ± 10.57	0.977	251.06 ± 11.36	**0.036**	242.25 ± 10.50	0.221	237.43 ± 11.74	0.568
	246.33 ± 5.94		246.42 ± 10.71		239.20 ± 6.09		234.20 ± 2.58		234.33 ± 10.25	
inner superior	250.12 ± 8.09	0.787	255.00 ± 24.93	0.750	253.11 ± 23.62	0.179	240.25 ± 3.50	0.539	240.86 ± 10.18	0.317
	251.08 ± 7.70		253.75 ± 25.36		242.60 ± 15.50		238.20 ± 6.68		246.67 ± 16.23	
outer superior	250.88 ± 6.39	0.367	257.67 ± 18.63	0.543	260.44 ± 22.89	**0.048**	246.25 ± 8.99	0.268	245.57 ± 15.61	0.775
	250.21 ± 6.37		255.33 ± 20.02		244.40 ± 7.43		238.20 ± 6.18		249.00 ± 10.67	
inner nasal	253.32 ± 6.44	0.880	256.00 ± 14.07	0.728	254.39 ± 17.03	0.191	251.75 ± 7.67	0.176	247.71 ± 17.28	0.431
	253.21 ± 6.93		253.83 ± 15.81		245.40 ± 9.39		244.60 ± 7.63		244.83 ± 15.80	
outer nasal	249.00 ± 6.29	0.521	253.50 ± 15.90	0.907	255.33 ± 18.95	0.080	246.75 ± 6.13	0.059	250.00 ± 8.22	0.520
	248.00 ± 6.03		252.67 ± 16.90		242.20 ± 9.09		238.00 ± 5.61		242.00 ± 10.90	
inner temporal	253.20 ± 10.02	0.976	256.58 ± 29.11	0.750	253.61 ± 22.14	0.232	243.50 ± 6.13	0.327	248.29 ± 13.62	0.250
	253.04 ± 10.20		252.67 ± 16.90		244.20 ± 8.10		239.40 ± 5.45		247.33 ± 14.41	
outer temporal	249.96 ± 8.57	0.560	258.58 ± 26.94	0.729	259.78 ± 24.89	0.117	249.25 ± 5.37	**0.036**	252.00 ± 16.41	0.943
	248.38 ± 7.42		259.42 ± 26.63		246.20 ± 10.08		242.20 ± 2.86		250.67 ± 12.43	
total volume	1.97 ± 0.07	**<0.001** **^#^**	1.81 ± 0.11	0.064	1.81 ± 0.11	0.156	1.74 ± 0.03	0.138	1.71 ± 0.16	0.720
	1.71 ± 0.36		1.73 ± 0.05		1.74 ± 0.06		1.71 ± 0.36		1.75 ± 0.08	
**RNFL Thickness**										
global	46.16 ± 4.36	0.880	41.45 ± 3.53	0.974	43.15 ± 9.53	0.414	41.75 ± 5.12	0.532	40.86 ± 4.22	0.474
	46.00 ± 4.40		42.36 ± 6.26		44.20 ± 3.27		39.80 ± 2.77		40.17 ± 8.65	
inferior temporal	48.72 ± 11.03	0.764	40.36 ± 7.24	0.666	41.80 ± 12.46	**0.563**	35.75 ± 11.26	0.219	38.29 ± 4.46	0.283
	46.75 ± 6.52		42.73 ± 7.86		45.00 ± 6.96		36.80 ± 4.65		44.67 ± 12.12	
inferior nasal	48.40 ± 8.23	0.666	45.45 ± 5.57	0.691	45.10 ± 9.33	0.134	41.00 ± 13.54	0.537	42.43 ± 8.77	0.774
	46.96 ± 6.19		44.82 ± 5.87		51.40 ± 6.80		40.20 ± 5.58		46.17 ± 10.77	
superior temporal	48.32 ± 8.82	0.645	43.73 ± 11.85	0.598	43.60 ± 22.02	0.454	40.50 ± 7.32	0.461	40.00 ± 12.66	0.721
	49.38 ± 8.15		40.64 ± 11.73		44.00 ± 9.43		37.40 ± 4.09		41.00 ± 12.61	
superior nasal	38.56 ± 9.40	0.952	36.00 ± 14.58	0.307	33.55 ± 22.62	0.324	34.00 ± 2.16	0.624	35.57 ± 8.65	0.317
	39.00 ± 8.20		32.64 ± 16.72		35.40 ± 4.03		36.80 ± 7.53		29.50 ± 9.99	
nasal	43.68 ± 7.12	0.968	41.82 ± 3.40	0.894	43.15 ± 8.52	0.973	45.25 ± 6.29	0.221	43.29 ± 3.54	0.719
	43.54 ± 6.15		42.82 ± 6.33		41.80 ± 6.90		40.60 ± 5.94		38.67 ± 12.06	
temporal	48.80 ± 8.41	0.833	41.18 ± 7.11	0.156	47.40 ± 12.04	0.539	45.75 ± 10.24	0.806	42.43 ± 8.03	0.943
	49.21 ± 8.21		46.18 ± 9.71		47.20 ± 6.09		42.60 ± 5.89		41.83 ± 8.32	
**GCL Thickness**										
central	22.32 ± 2.30	0.951	18.92 ± 3.47	0.663	16.78 ± 3.59	0.107	15.25 ± 1.50	0.080	15.57 ± 3.78	0.195
	22.46 ± 2.10		19.42 ± 3.23		19.40 ± 2.70		18.20 ± 2.49		17.67 ± 3.55	
inner inferior	27.88 ± 2.10	0.610	26.25 ± 2.05	0.741	23.50 ± 4.79	0.650	24.00 ± 1.41	0.167	21.57 ± 5.59	0.109
	28.29 ± 1.57		26.33 ± 2.10		25.20 ± 1.64		25.40 ± 1.51		24.17 ± 3.25	
outer inferior	26.84 ± 1.77	0.575	26.25 ± 3.59	0.497	24.44 ± 4.09	0.218	23.25 ± 2.98	0.366	22.14 ± 6.28	0.560
	27.25 ± 1.35		26.58 ± 3.50		24.40 ± 1.67		24.80 ± 0.83		22.17 ± 4.79	
inner superior	26.40 ± 1.58	0.324	22.42 ± 4.18	0.680	21.50 ± 4.50	0.142	18.00 ± 2.44	**0.044**	17.57 ± 4.92	0.150
	26.96 ± 1.92		22.08 ± 3.82		23.60 ± 3.64		21.20 ± 1.09		21.17 ± 3.97	
outer superior	25.40 ± 2.50	0.647	23.50 ± 3.89	0.520	22.44 ± 3.60	0.228	24.75 ± 2.21	0.900	20.14 ± 5.64	0.387
	25.71 ± 2.57		22.83 ± 3.71		24.20 ± 2.77		24.80 ± 0.83		22.00 ± 5.21	
inner nasal	26.16 ± 2.35	0.723	23.50 ± 1.78	0.379	21.56 ± 5.05	0.410	19.00 ± 4.08	0.082	19.57 ± 5.31	0.221
	26.38 ± 2.33		22.83 ± 2.03		23.80 ± 2.38		22.60 ± 1.81		21.67 ± 5.78	
outer nasal	26.68 ± 1.72	0.569	25.50 ± 1.62	0.479	22.00 ± 6.29	0.499	22.25 ± 4.99	0.530	22.00 ± 1.78	0.685
	26.96 ± 1.75		24.92 ± 2.02		24.40 ± 2.07		24.80 ± 0.83		21.50 ± 3.93	
inner temporal	25.56 ± 3.34	0.520	23.50 ± 4.70	0.748	20.89 ± 5.31	**0.032**	20.25 ± 1.70	0.268	18.71 ± 4.92	0.425
	26.33 ± 2.18		23.08 ± 4.83		23.80 ± 1.92		21.60 ± 4.61		21.50 ± 4.18	
outer temporal	26.72 ± 2.57	0.349	25.42 ± 3.47	0.605	25.00 ± 5.16	0.733	25.00 ± 1.41	0.898	20.43 ± 5.28	0.278
	27.46 ± 1.56		25.17 ± 3.38		24.80 ± 2.86		24.00 ± 2.82		22.50 ± 3.14	
total volume	0.18 ± 0.01	0.668	0.16 ± 0.00	0.914	0.16 ± 0.02	0.259	0.15 ± 0.01	0.093	0.13 ± 0.03	0.343
	0.19 ± 0.01		0.17 ± 0.01		0.17 ± 0.01		0.17 ± 0.01		0.15 ± 0.03	

Abbreviations: RE: right eye; RNFL: retinal nerve fiber layer; GCL: ganglion cell layer; thickness in microns (μm); mean ± SD (SD: standard deviation); *p* < 0.05 statistical significance; *p* < 0.02; ^#^ statistical significance with Bonferroni correction for multiple comparisons; w: week; retinal thickening shown in light gray and GCL thinning under the episcleral vein sclerosis model shown in dark gray.

**Table 3 biomedicines-09-00682-t003:** Neuroretinal loss rate measured by optical coherence tomography in both ocular hypertensive models (episcleral and microspheres).

All-Sector Average Loss Rate (nm)/mmHg/day												
Right Eye							Left Eye					
Time	RETINA		RNFL		GCL		RETINA		RNFL		GCL	
	EPIm	Ms20/10	EPIm	Ms20/10	EPIm	Ms20/10	EPIm	Ms20/10	EPIm	Ms20/10	EPIm	Ms20/10
8 w	2	4	−5	−8	−2	−5	−7	−31	−4	−18	−2	−12
12 w	4	−6	−5	−2	−5	−3	−5	−6	−10	−1	−8	−4
18 w	−8	−11	−7	−8	−6	−4	−12	−14	−9	−9	−8	−3
24 w	−3	−2	−3	−2	−3	−2	−3	−2	−3	−2	−2	−2
Average	−1	−4	−5	−5	−4	−3	−7	−14	−7	−7	−5	−5

Abbreviations: EPIm: episcleral sclerosis model; Ms 20/10: microsphere 20/10 model; RNFL: retinal nerve fiber layer; GCL: ganglion cell layer; thickness in nanometers (nm); w: week.

## Data Availability

Data supporting the findings of this study are available from the corresponding author [E.G.-M.] on request.

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
