# Peer review of "Chronic Glaucoma Using Biodegradable Microspheres to Induce Intraocular Pressure Elevation. Six-Month Follow-Up"

_biomedicines, 2021, doi:10.3390/biomedicines9060682_

Round 1

Reviewer 1 Report

Comments are provided in the attached file.

Reviewer 2 Report

The authors compared two rat models for glaucoma. This study is well designed and conducted. I only have a few comments:

  1. Bonferroni correction is probably too harsh as the those comparisons are not totally random. False discovery rate correction probably can give better results.
  2. Maybe the authors can further perform survival analyses with some thresholds to define glaucoma or ocular hypertension.
  3. If the authors could further test and compare the rat's response to light stimuli between the two models, the models will be more convincing.
  4. Three decimal digits precision is unnecessary. In most cases, you only need two decimal digits.

Round 2

Reviewer 1 Report

The authors have addressed all the review questions